# ViSPLA: Visual Iterative Self-Prompting for Language-Guided 3D Affordance Learning

**Hritam Basak**
Department of Computer Science
Stony Brook University
Stony Brook, NY, USA
hbasak@cs.stonybrook.edu

**Zhaozheng Yin**
Department of Computer Science
Stony Brook University
Stony Brook, NY, USA
zyin@cs.stonybrook.edu

## Abstract

We address the problem of language-guided 3D affordance prediction, a core capability for embodied agents interacting with unstructured environments. Existing methods often rely on fixed affordance categories or require external expert prompts, limiting their ability to generalize across different objects and interpret multi-step instructions. In this work, we introduce *ViSPLA*, a novel iterative self-prompting framework that leverages the intrinsic geometry of predicted masks for continual refinement. We redefine affordance detection as a language-conditioned segmentation task: given a 3D point cloud and language instruction, our model predicts a sequence of refined affordance masks, each guided by differential geometric feedback including Laplacians, normal derivatives, and curvature fields. This feedback is encoded into visual prompts that drive a multi-stage refinement decoder, enabling the model to self-correct and adapt to complex spatial structures. To further enhance precision and coherence, we introduce Implicit Neural Affordance Fields, which define continuous probabilistic regions over the 3D surface without additional supervision. Additionally, our Spectral Convolutional Self-Prompting module operates in the frequency domain of the point cloud, enabling multi-scale refinement that captures both coarse and fine affordance structures. Extensive experiments demonstrate that *ViSPLA* achieves state-of-the-art results on both seen and unseen objects on two benchmark datasets. Our framework establishes a new paradigm for open-world 3D affordance reasoning by unifying language comprehension with low-level geometric perception through iterative refinement. [Project Website](#)

## 1 Introduction

Affordance, initially conceptualized by Gibson [1], defines the potential action possibilities that objects present to an agent. The evolution of robotic systems toward increasingly unstructured environments necessitates a fundamental paradigm shift in how we conceptualize affordance detection. Formally, we can represent the affordance detection problem as a mapping function $f_\theta : (\mathcal{P}) \mapsto \mathcal{A}$, where $\mathcal{P} \in \mathbb{R}^{N \times 3}$ denotes a point cloud with $N$ points, and $\mathcal{A}$ is the binary affordance mask indicating interactable regions. As shown in Figure 1(a), traditional approaches constrain this mapping to a limited set of predefined $K$ affordance categories $\mathcal{A} = \{a_k\}; k = \{1, 2, .., K\}$, which fundamentally restricts the generalization capability and operational flexibility in dynamic, real-world environments [2]. Although conventional methodologies have predominantly focused on visual modalities, attempting to infer functionality from geometric structures or 2D visual features, such approaches inherently lack the semantic reasoning capabilities essential for complex interaction scenarios. The semantic gap between low-level perceptual features and high-level functional understanding represents a critical

39th Conference on Neural Information Processing Systems (NeurIPS 2025).

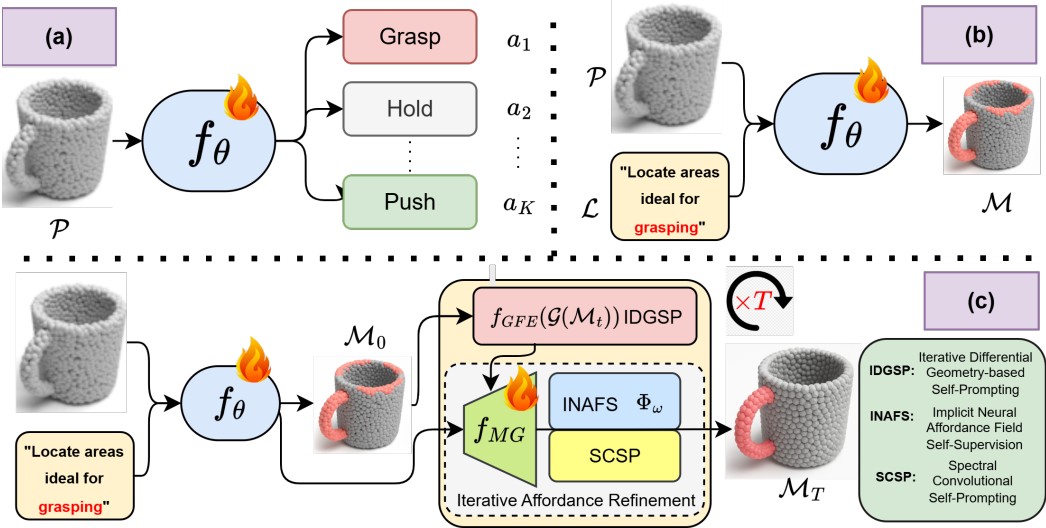

Figure 1: (a) Traditional vision-based methods [3, 4] rely on trainable network $f_\theta$ to predict a fixed set of affordances $f_\theta : \mathcal{P} \mapsto \mathcal{A}$; $\mathcal{A} = \{a_1, a_2, ..a_K\}$; (b) Language input along with point-cloud add more flexibility to comprehend complex language instructions and mitigate the problem of open-set affordance prediction [5, 6]: $f_\theta : (\mathcal{P}, \mathcal{L}) \mapsto \mathcal{M}$; (c) We propose refining the initial affordance prediction $\mathcal{M}_0$ for $T$ steps using our proposed IDGSP, and Iterative Affordance Refinement module, consisting of multi-stage refinement decoder $f_{MG}$, INAFS ($\Phi_\omega$) and SCSP. Our solution could be formulated as $f_\theta : (\mathcal{P}, \mathcal{G}(\mathcal{M}_{t-1}), \mathcal{L}) \mapsto \mathcal{M}_t$; $t \in \{1, 2, ..T\}$. Details can be found in section 3.

limitation that inhibits the deployment of autonomous agents in real-world contexts. Language-guided affordance prediction offers a mathematically elegant solution to this complex problem.

Language-guided affordance detection from 3D point clouds represents a pivotal direction in embodied AI, serving as the critical bridge between perception and manipulation in the physical world. By conditioning the affordance function on natural language instructions, we can formulate a more generalizable mapping: $\mathcal{F}_\theta : (\mathcal{P}, \mathcal{L}) \mapsto \mathcal{M}$, where $\theta$ represents the learnable parameters, $\mathcal{L}$ represents a linguistic instruction, and $\mathcal{M} \in \{0, 1\}^N$ is the binary affordance mask (visualized in Figure 1(b)). This formulation opens avenues for handling more diverse and complex scenarios—potentially allowing models to interpret novel affordance types via linguistic cues, handle multi-step tasks through decomposed predictions, and relate instructions to affordances at different levels of granularity. Recent progress in Large Language Models (LLMs) has shown impressive capabilities in sequential reasoning and knowledge grounding [7], but these models are often decoupled from 3D perception. Meanwhile, 3D affordance detection methods typically remain limited to static, single-affordance settings, with little capacity to handle instructions requiring compositional or context-aware reasoning across multiple object parts. This disconnect motivates a more integrated, multimodal approach that unifies linguistic understanding with spatial perception.

To this end, we propose an iterative self-prompting-based 3D affordance detection paradigm that bridges the gap between language understanding and affordance segmentation through geometric feedback-driven refinement, as shown in Figure 1(c). Unlike prior approaches that perform single-pass inference, our method implements a closed-loop system where each predicted affordance mask is used to generate geometric self-prompts that refine subsequent predictions. Mathematically, we formulate this as: $\mathcal{M}_t = f_\theta\big(\mathcal{P}, \mathcal{G}(\mathcal{M}_{t-1}), \mathcal{L}\big)$; $t \in \{1, 2, .., T\}$, where $\mathcal{M}_0 = f_\theta(\mathcal{P}, \mathcal{L})$ is the initial affordance mask predicted from a language-conditioned decoder, and $\mathcal{G}$ denotes the geometric prompt generator that extracts differential features (e.g., curvature, normal derivatives) from $\mathcal{M}_{t-1}$. The final refined mask $\mathcal{M}_T$ integrates both semantic guidance and geometric consistency, enabling robust and generalizable affordance segmentation across varying levels of granularity and complexity.

This approach addresses several critical challenges in the field: **(1)** Existing single-pass inference methods lack the ability to iteratively refine predictions, often leading to suboptimal segmentation, especially on complex geometries; **(2)** most affordance models fail to leverage intrinsic geometric structure for mask refinement, relying instead on language cues alone, which limits localization

accuracy, especially in complex or ambiguous settings; **(3)** the disconnect between high-level language semantics and low-level geometric features, hindering precise and context-aware affordance prediction across multiple scales; and **(4)** the difficulty of achieving fine-grained, geometrically consistent affordance boundaries without dense supervision, particularly in sparse or noisy point clouds. In summary, our contributions are:

- We introduce Visual Iterative Self-Prompting for 3D Affordance Learning (ViSPLA), which leverages geometric features from predicted masks as visual prompts for progressive refinement. Unlike existing single-pass methods, our approach establishes a self-improving cycle that enhances precision across multiple object geometries.
- We propose a novel Differential Geometric Self-Prompting mechanism that extracts mathematical properties (Laplacians, curvatures, normal derivatives) from predicted masks to guide subsequent iterations. This approach enables more accurate affordance localization by incorporating intrinsic geometric cues rather than relying solely on language.
- We develop a Multi-Stage Refinement Decoder that creates dynamic mappings between language tokens and point features. By injecting LLM reasoning into dense point features, our approach bridges high-level semantic understanding with low-level geometric representation.
- We introduce an Implicit Neural Affordance Field technique that learns a smooth, continuous function over the 3D object to refine affordance boundaries and enforce geometric consistency, even without extra supervision. In tandem, our Spectral Convolutional Self-Prompting module analyzes and enhances affordance predictions at multiple structural scales, enabling the model to capture both broad shapes and fine details for robust and accurate segmentation, especially in sparse or noisy scenarios.
- We demonstrate that fine-tuning pre-trained MLLMs through our self-prompting framework yields superior performance on both seen and unseen scenarios.

## 2 Related Work

### 2.1 Affordance Detection

Affordance detection aims to identify functionally interactive regions on objects, crucial for enabling robotic agents to manipulate and reason about the physical world. Early works in 2D explored object-level affordances using CNNs [3], later extending to language-conditioned queries [8], but remained limited to coarse spatial reasoning. Subsequent efforts [9, 10] introduced fine-grained part-level detection but were restricted to fixed affordance taxonomies. The emergence of 3D datasets like 3D AffordanceNet [4] and PartNet [11] enabled point cloud-based affordance learning. IAGNet [12] utilized 2D human-object interactions to guide 3D segmentation, while OpenAD [13] advanced open-vocabulary affordance detection using joint text-geometry embeddings. However, these models still rely on static label spaces and do not support complex instruction understanding. Recent methods like LASO [14] incorporate language into 3D affordance prediction, but often assume one-to-one mappings between text and affordance, lacking support for multi-step or compositional reasoning. Chu *et al.* [15] use LLMs for cross-modal object retrieval but cannot produce spatially grounded interaction masks.

In contrast, our work formulates affordance detection as an instruction-conditioned segmentation task that enables open-vocabulary, multi-step reasoning directly over 3D point clouds, overcoming the rigidity of fixed labels and the limitations of prior semantic alignment methods.

### 2.2 Multimodal Large Language Models

Multimodal Large Language Models (MLLMs) extend the language understanding capabilities of LLMs to the visual and spatial domains [16, 17] by aligning textual tokens with visual and geometric inputs. Initial breakthroughs in 2D MLLMs [18–20] enabled joint reasoning over images and text, yet these models lacked the granularity necessary for fine-grained visual tasks such as segmentation. To mitigate this, models like LLaVA [21] introduced spatial localization, improving regional understanding.

Inspired by this progress, researchers have begun extending MLLMs to the 3D domain. Object-centric MLLMs such as PointLLM [22] and ShapeLLM [23] utilize point-based encoders and

multi-view distillation to encode geometric structure and semantics. These models demonstrate strong performance in 3D captioning and object-level referring expression grounding, but often operate on isolated objects and fail to model complex inter-object spatial dependencies. Scene-level LMMs such as Chat-3D [24], LL3DA [25], and 3D-LLM [26] extend grounding to richer indoor scenes using object identifiers, positional embeddings, and pre-selection modules to facilitate dialogue-driven scene understanding.

However, despite these advances, existing 3D MLLMs predominantly focus on global grounding and object identification, lacking the capacity for localized, affordance-specific segmentation or functional reasoning over object parts. 3D-AffordanceLLM [6] takes a step forward by introducing an <AFF> token and a custom decoder to generate affordance masks from natural language queries. Unlike [6], which performs single-pass instruction-to-mask mapping with no feedback loop, our method introduces an Iterative Self-Prompting mechanism that progressively refines predictions by leveraging prior affordance masks as feedback prompts. Moreover, 3D-AffordanceLLM lacks any geometric introspection; in contrast, our Differential Geometric Self-Prompting explicitly uses curvature, Laplacian, and boundary topology cues for precise localization, going beyond language-only guidance. While 3D-AffordanceLLM relies on a fixed decoder architecture, our Multi-scale Visual-Language Integration Module dynamically aligns instructions with geometric features at varying resolutions, enhancing affordance prediction across objects of diverse scale and complexity. Finally, unlike their static segmentation output, our Affordance Dictionary Adaptive Fusion fuses temporal and functional context across steps, enabling robust multi-stage affordance reasoning. Together, these advances allow our model to achieve superior open-world generalization and performance in sequential, instruction-grounded 3D affordance tasks.

## 3 Proposed Method

We propose *ViSPLA*, a novel iterative self-prompting framework for language-guided 3D affordance detection that incorporates differential geometric feedback for progressive mask refinement. Unlike previous methods that rely on single-pass inference, our approach employs a recurrent self-prompting mechanism that leverages the intrinsic geometric properties of predicted affordance masks to guide subsequent refinements.

### 3.1 Probelm Formulation

Following the paradigm reformation introduced by 3D-AffordanceLLM [6], we formulate affordance detection as an Instruction Reasoning Affordance Segmentation (IRAS) task. Given a natural language instruction $\mathcal{L}$ and a point cloud $\mathcal{P} \in \mathbb{R}^{N \times 3}$ containing $N$ points, our goal is to predict a binary affordance mask $\mathcal{M} \in \{0, 1\}^N$ indicating regions suitable for the specified interaction. While existing approaches [6] model this as a direct mapping $f_\theta : (\mathcal{P}, \mathcal{L}) \mapsto \mathcal{M}$, we introduce an iterative refinement process, as already described in section 1:

$$\mathcal{M}_t = f_\theta\big(\mathcal{P}, \mathcal{G}(\mathcal{M}_{t-1}), \mathcal{L}\big);\ t \in \{1, 2, ..., T\} \tag{1}$$

where $\mathcal{M}_0 = f_\theta(\mathcal{P}, \mathcal{L})$ is the initial affordance prediction and $\mathcal{G}$ is our proposed geometric prompt generator that extracts meaningful differential features from previous mask predictions. *ViSPLA* consists of three main components: **(1)** a language-guided affordance detection backbone based on 3D-AffordanceLLM, **(2)** a differential geometry-based self-prompting module, and **(3)** an iterative affordance refinement module, consisting of a multi-stage refinement decoder, implicit neural field supervision, and spectral convolutional self-prompting. The overall workflow is shown in Figure 2.

### 3.2 Preliminaries: Language-guided Affordance Detection Backbone

We build upon the 3D-AffordanceLLM [6] architecture, adopting it as our backbone, which comprises a pre-trained point encoder $f_{\mathrm{PE}}$, a point cloud backbone $f_{\mathrm{PB}}$, a projection module $f_{\mathrm{proj}}$, a large language model $f_{\mathrm{LLM}}$, and an affordance decoder $f_{\mathrm{AFD}}$. Given an input point cloud $\mathcal{P}$ and a natural language instruction $\mathcal{L}$, the system proceeds as follows: the point encoder first extracts geometric features $X = f_{\mathrm{PE}}(\mathcal{P}) \in \mathbb{R}^{m \times c}$, where $m$ denotes the number of keypoints and $c$ is the feature dimension. These features are projected into the token space via $f_{\mathrm{proj}}$, yielding $Y = f_{\mathrm{proj}}(X) \in \mathbb{R}^{m \times d}$, where $d$ matches the dimensionality of the LLM token embeddings. The projected point tokens are concatenated with the instruction tokens and passed into the LLM $f_{\mathrm{LLM}}$, which processes them

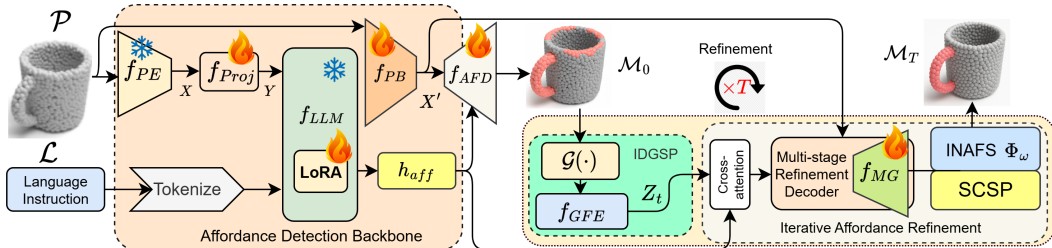

Figure 2: Overview of the *ViSPLA* framework: given a point cloud $\mathcal{P}$ and a language instruction $\mathcal{L}$, first we extract geometric features $X = f_{PE}(\mathcal{P})$, project them to $Y = f_{proj}(X)$, and pass them along with language tokens to the frozen LLM with trainable LoRA layer. Next, dense point-cloud features $X' = f_{PB}(\mathcal{P})$ and affordance tokens from LLM $h_{aff} = f_{LLM}(Y, \mathcal{L})$ are extracted and passed to the affordance decoder $f_{AFD}$ to produce an initial mask $\mathcal{M}_0$. The iterative affordance refinement module then refines the mask via $T$ steps. At each iteration $t$, geometric descriptors $Z_t = f_{GFE}(\mathcal{G}(\mathcal{M}_{t-1}))$ (e.g., Laplacian, curvature) are computed (subsection 3.3) and injected as visual prompts, along with $X'$ and $h_{aff}$ to multi-stage refinement decoder $f_{MG}$ (subsubsection 3.4.1) to produce the refined affordance prediction $\mathcal{M}_t$, after processing them through INAFS (subsubsection 3.4.2) and SCSP (subsubsection 3.4.3). The process converges after $T$ steps, enabling precise, language-guided 3D affordance segmentation $\mathcal{M}_T$ through closed-loop geometric feedback.

to generate a response sequence that includes a special affordance token <AFF>. The hidden representation corresponding to <AFF>, denoted as $h_{\text{aff}}$, is then extracted. Meanwhile, the point cloud backbone $f_{\text{PB}}$ computes dense point-wise features $X' = f_{\text{PB}}(\mathcal{P}) \in \mathbb{R}^{N \times c'}$. Finally, the affordance decoder $f_{\text{AFD}}$ fuses $h_{\text{aff}}$ with $X'$ to produce the initial affordance mask $\mathcal{M}_0 = f_{\text{AFD}}(h_{\text{aff}}, X')$.

### 3.3 Iterative Differential Geometry-based Self-Prompting

Building upon the initial prediction $\mathcal{M}_0$, we introduce our core contribution: *Iterative Differential Geometry-Based Self-Prompting (IDGSP)*. This module leverages geometric feedback derived from prior affordance masks to progressively refine segmentation outputs. At each iteration $t$, we compute a set of differential geometric descriptors from the previous mask $\mathcal{M}_{t-1}$:

$$\mathcal{G}(\mathcal{M}_{t-1}) = \left\{ \nabla^2 \mathcal{M}_{t-1}, \nabla \mathcal{M}_{t-1} \cdot \mathbf{n}, \mathcal{H}(\mathcal{M}_{t-1}), \kappa_1(\mathcal{M}_{t-1}), \kappa_2(\mathcal{M}_{t-1}) \right\} \tag{2}$$

where $\nabla^2 \mathcal{M}_{t-1}$ denotes the Laplacian of the mask capturing local curvature variation, $\nabla \mathcal{M}_{t-1} \cdot \mathbf{n}$ represents the normal derivative quantifying alignment with surface normals, $\mathcal{H}(\mathcal{M}_{t-1})$ is the mean curvature of mask boundaries, and $\kappa_1, \kappa_2$ are the principal curvatures. These geometric signals encode essential boundary-aware and topological properties that reflect the physical plausibility of affordance regions. The extracted descriptors are transformed into a dense per-point representation $\mathcal{Z}_t \in \mathbb{R}^{N \times d}$ using a learnable geometric feature extractor $f_{\text{GFE}}$: $\mathcal{Z}_t = f_{\text{GFE}}(\mathcal{G}(\mathcal{M}_{t-1}))$. This geometry-driven prompt $\mathcal{Z}_t$ is then injected into the multi-stage refinement decoder (subsection 3.4.1) to guide the refinement process.

### 3.4 Iterative Affordance Refinement

#### 3.4.1 Multi-Stage Refinement Decoder

To operationalize the geometric self-prompting mechanism, we employ a multi-stage refinement decoder that iteratively updates the affordance mask using both language and geometry-informed cues. At each iteration $t$, the geometric features $\mathcal{Z}_t$ are fused with the initial LLM embedding $h_{\text{aff}}$ using a cross-attention mechanism:

$$h_{\text{aff}}^{(t)} = \text{CrossAttn}(h_{\text{aff}}, \mathcal{Z}_t) \tag{3}$$

The refined embedding $h_{\text{aff}}^{(t)}$ is then combined with dense point cloud features $X'$ via a mask generation module $f_{\text{MG}}$ to produce the updated affordance mask:

$$\mathcal{M}_t = f_{\text{MG}}(h_{\text{aff}}^{(t)}, X') \tag{4}$$

### 3.4.2 Implicit Neural Affordance Field Self-Supervision

To complement the discrete iterative refinement process with a smooth, continuous representation, we incorporate a regularization strategy based on implicit neural fields to enhance boundary precision and geometric consistency without relying on additional labels. This component learns a continuous implicit function $\Phi_\omega : \mathbb{R}^3 \times \mathbb{R}^d \to [0, 1]$, parameterized by $\omega$, which maps any 3D point $\mathbf{x} \in \mathbb{R}^3$ and its corresponding feature vector to a scalar-valued affordance probability.

The function $\Phi_\omega$ is trained via energy minimization loss $\mathcal{L}_{INAFS} = \mathcal{E}(\Phi_\omega)$ over the 3D spatial domain $\Omega$, incorporating geometric priors and alignment with the current mask predictions. The energy term is defined as:

$$\mathcal{E}(\Phi_\omega) = \int_\Omega \|\nabla \Phi_\omega(\mathbf{x})\|^2 d\mathbf{x} + \lambda_1 \int_{\partial\Omega} \left(\Phi_\omega(\mathbf{x}) - \mathcal{M}(\mathbf{x})\right)^2 d\mathbf{x} + \lambda_2 \int_\Omega \left(|\Delta\Phi_\omega(\mathbf{x})| - \beta\|\kappa(\mathbf{x})\|\right)^2 d\mathbf{x}$$

(5)

Here, the first term encourages spatial smoothness by minimizing the gradient norm of the implicit field. The second term enforces fidelity to the current predicted mask $\mathcal{M}(\mathbf{x})$ at the boundary $\partial\Omega$, ensuring consistency with previously inferred affordances. The third term aligns the second-order variation of the field, measured by the Laplacian $\Delta\Phi_\omega$, with the Gaussian curvature $\kappa(\mathbf{x})$ (where $\kappa = \kappa_1 \cdot \kappa_2$), thereby promoting geometric conformity with intrinsic surface structures. The weighting parameters $\lambda_1$, $\lambda_2$, and scaling constant $\beta$ balance the contributions of fidelity and curvature alignment. After optimization, the final affordance mask is extracted by thresholding the implicit field at 0.5:

$$\mathcal{M}_{\text{refined}} = \{\mathbf{x} \in \mathcal{P} \mid \Phi_\omega(\mathbf{x}) > 0.5\}$$

(6)

This implicit representation allows the model to refine coarse predictions into geometrically consistent and semantically plausible affordance regions, even in the absence of explicit supervision.

### 3.4.3 Spectral Convolutional Self-Prompting

To complement spatial refinement with a frequency-aware perspective, we introduce *Spectral Convolutional Self-Prompting (SCSP)*, which enables the model to capture affordance structures at multiple scales in the spectral domain of point cloud. We treat the 3D point cloud as a discrete manifold encoded by the normalized Laplacian operator $\mathbf{L} = \mathbf{I} - \mathbf{D}^{-1/2}\mathbf{A}\mathbf{D}^{-1/2}$, where $\mathbf{A}$ is the affinity matrix derived from local geometric similarity, and $\mathbf{D}$ is the corresponding degree matrix. Given the predicted affordance mask $\mathcal{M} \in \mathbb{R}^N$, we project it to spectral domain via eigen-decomposition:

$$\hat{\mathcal{M}} = \sum_{i=1}^{N} \alpha_i \mathbf{u}_i, \quad \text{where} \quad \alpha_i = \langle \mathcal{M}, \mathbf{u}_i \rangle$$

(7)

Here, $\{\mathbf{u}_i\}_{i=1}^N$ are the eigenvectors of $\mathbf{L}$, and $\{\alpha_i\}$ are the corresponding spectral coefficients. Refinement is performed by applying a learnable spectral filter $g(\lambda_i)$, parameterized over the eigenvalues $\{\lambda_i\}$, yielding the updated mask in the spectral domain:

$$\hat{\mathcal{M}}_{t+1} = \sum_{i=1}^{N} g(\lambda_i)\alpha_i^{(t)}\mathbf{u}_i$$

(8)

By operating in the spectral domain, SCSP provides a principled, resolution-aware mechanism for affordance enhancement without explicit hierarchical supervision. The entire refinement process is performed iteratively for $T$ steps, allowing the model to progressively improve affordance localization by incorporating differential geometric feedback.

### 3.5 Overall Learning Strategy

To effectively address data scarcity and ensure robust affordance understanding, we adopt a multi-stage training strategy inspired by 3D-AffordanceLLM [6]. The pre-trained backbone is frozen, and we train the proposed self-prompting modules, including IDGSP, INAFS, and SCSP, to refine affordance masks using geometric and spectral cues. Our overall loss combines multitask objectives:

$$\mathcal{L} = \lambda_{\text{txt}}\mathcal{L}_{\text{txt}} + \lambda_{\text{mask}}\mathcal{L}_{\text{mask}} + \lambda_{\text{IDGSP}}\mathcal{L}_{\text{IDGSP}} + \lambda_{\text{INAFS}}\mathcal{L}_{\text{INAFS}} + \lambda_{\text{SCSP}}\mathcal{L}_{\text{SCSP}}$$

(9)

Here $\mathcal{L}_{\text{txt}}$ is autoregressive CE loss for LLM response generation, $\mathcal{L}_{\text{mask}}$ is BCE + Dice loss for initial affordance mask prediction, both used in the affordance backbone (following [6]). $\mathcal{L}_{\text{IDGSP}}$ penalizes

inconsistencies between iterative mask refinements and encourages smoothness, $\mathcal{L}_{\text{INAFS}}$ denotes energy-based regularization over the implicit affordance field (described in subsubsection 3.4.2), $\mathcal{L}_{\text{SCSP}}$ performs spectral consistency and spatial regularization using total variation $TV$. Specifically, $\mathcal{L}_{\text{IDGSP}}$ and $\mathcal{L}_{\text{SCSP}}$ are formulated as:

$$\mathcal{L}_{\text{IDGSP}} = \sum_{t=1}^{T} \lambda_t \|\mathcal{M}_t - \mathcal{M}_{t-1}\|_{W_{2,2}}^2 + \alpha \|\nabla^4 \mathcal{M}_T\|_2^2, \text{ and} \tag{10}$$

$$\mathcal{L}_{\text{SCSP}} = \sum_{t=1}^{T} \sum_{k=1}^{K} \gamma_k \|W_k(\hat{\mathcal{M}}_t - \hat{\mathcal{M}}_{t-1})\|_F^2 + \tau \text{TV}(\mathcal{M}_T) \tag{11}$$

where in Equation 10, $\| \cdot \|_{W_{2,2}}^2$ is the Sobolev $W^{2,2}$ norm measuring the difference between consecutive masks, capturing both value differences and derivatives (geometric properties), $\lambda_t$ is iteration-specific weight, $\nabla^4$ is biharmonic operator, and $\alpha |\nabla^4 \mathcal{M}_T|_2^2$ is Tikhonov regularization term ensuring the final mask has smooth boundaries with controlled curvature. The first term in Equation 11 penalizes changes in the spectral components of the mask across iterations, enforcing frequency-consistent refinement, whereas the second term ensures that the final predicted mask $\mathcal{M}_T$ is spatially smooth, reducing over-segmentation and promoting contiguous affordance regions. $\sum_{t=1}^{T}$ is summation over all iterations of the self-prompting process (from 1 to $T$), $\sum_{k=1}^{K}$ is summation over $K$ different frequency bands or scales of analysis, $W_k$ is diagonal matrix that isolates the $k$-th frequency band, $\gamma_k$ is scale-dependent weight coefficients for each frequency band $k$, $\hat{\mathcal{M}}_t$ is spectral decompositions of affordance masks at iteration $t$, $| \cdot |_F^2$ is Frobenius norm squared, measuring differences in frequency components, $\tau$ denotes weight parameter balancing spectral consistency and spatial coherence, and $\text{TV}(\mathcal{M}_T)$ is total variation regularizer promoting spatial smoothness in the final mask.

This multi-stage optimization pipeline enables the model to progressively refine affordance predictions by integrating linguistic reasoning with geometric and spectral feedback, leading to improved generalization and mask accuracy in open-world settings.

## 4 Experiments and Results

### 4.1 Dataset Description

Following previous works [5, 14], we conduct evaluations on two complementary 3D affordance datasets: PIAD [12] and LASO [14], each designed to test different aspects of generalization. PIAD serves as a complementary benchmark, comprising $7,012$ point clouds from the same object categories as LASO but introduces a stricter generalization setting—entire object instances are withheld from training, requiring the model to predict affordances on previously unseen geometries. As PIAD lacks textual annotations, we augment it with language instructions by sampling prompts from LASO's question pool, ensuring semantic alignment with each target affordance type. This design enables evaluation of our model's robustness in both instruction-conditioned and shape-driven generalization scenarios. LASO, on the other hand, contains $19,751$ language-guided point cloud pairs spanning $8,434$ unique object instances across 23 object categories and 17 affordance types. It supports both Seen and Unseen splits, where the Unseen configuration deliberately excludes specific affordance-object combinations during training to assess zero-shot generalization.

### 4.2 Implementation Details

Following 3D-AffordanceLLM [6], we utilize Phi-3.5-mini-instruct [27] as our base LLM with LoRA [28] fine-tuning. For 3D processing, we adopt Point-BERT [29] pre-trained with ULIP-2 [30] as our point encoder ($f_{PE}$) and Point Transformer [31] as our point backbone ($f_{PB}$). The feature dimension $d$ is set to 512 for both language and point features. The projector layer ($f_{proj}$) is implemented as a simple linear layer

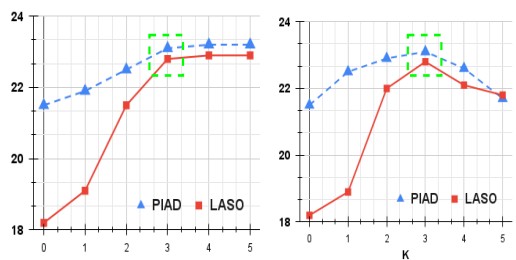

Figure 3: Performance analysis (aIoU on "seen" setting) with varying $T$ and $K$ values.

Table 1: Qualitative comparison of our proposed method on the PIAD (left) and LASO (right) datasets. The best and second-best results are highlighted in red and blue, respectively. LASO* indicates reported results of LASO [14] in GEAL [5]. † denotes our reproduced results of [6].

| Type | Method | aIoU ↑ | AUC ↑ | SIM ↑ | MAE ↓ |
|---|---|---|---|---|---|
| Seen | MBDF [33] | 9.3 | 74.9 | 0.415 | 0.143 |
| | PMF [34] | 10.1 | 75.1 | 0.425 | 0.141 |
| | FRCNN [35] | 12.0 | 76.1 | 0.429 | 0.136 |
| | ILN [36] | 11.5 | 75.8 | 0.427 | 0.137 |
| | PFusion [37] | 12.3 | 77.5 | 0.432 | 0.135 |
| | XMF [38] | 12.9 | 78.2 | 0.441 | 0.127 |
| | IAGNet [12] | 20.5 | 84.9 | 0.545 | 0.098 |
| | LASO [14] | 19.7 | 84.2 | 0.590 | 0.096 |
| | 3DAffLLM† [6] | 21.5 | 82.6 | 0.643 | 0.104 |
| | GEAL [5] | 22.5 | 85.0 | 0.601 | 0.092 |
| | **Ours** | **23.1** | **85.8** | **0.664** | **0.089** |
| Unseen | MBDF [33] | 4.2 | 58.2 | 0.325 | 0.213 |
| | PMF [34] | 4.7 | 60.3 | 0.330 | 0.211 |
| | FRCNN [35] | 5.1 | 61.9 | 0.332 | 0.195 |
| | ILN [36] | 4.7 | 59.7 | 0.325 | 0.207 |
| | PFusion [37] | 5.3 | 61.9 | 0.33 | 0.193 |
| | XMF [38] | 5.7 | 62.6 | 0.342 | 0.186 |
| | IAGNet [12] | 8.0 | 71.8 | 0.352 | 0.127 |
| | LASO [14] | 8.0 | 69.2 | 0.386 | 0.118 |
| | 3DAffLLM† [6] | 7.4 | 71.0 | 0.413 | 0.115 |
| | GEAL [5] | 8.7 | 72.5 | 0.390 | 0.102 |
| | **Ours** | **9.2** | **73.1** | **0.431** | **0.099** |

| Type | Method | aIoU ↑ | AUC ↑ | SIM ↑ | MAE ↓ |
|---|---|---|---|---|---|
| Seen | ReferTrans [39] | 13.7 | 79.8 | 0.497 | 0.124 |
| | ReLA [40] | 15.2 | 78.9 | 0.532 | 0.118 |
| | 3D-SPS [41] | 11.4 | 76.2 | 0.433 | 0.138 |
| | IAGNet [12] | 17.8 | 82.3 | 0.561 | 0.109 |
| | LASO [14] | 20.8 | 87.3 | 0.629 | 0.093 |
| | LASO* [14] | 19.7 | 85.2 | 0.600 | 0.097 |
| | 3DAffLLM† [6] | 18.2 | 84.9 | 0.622 | 0.104 |
| | GEAL [5] | 22.0 | 86.7 | 0.634 | 0.092 |
| | **Ours** | **22.8** | **87.3** | **0.651** | **0.090** |
| Unseen | ReferTrans [39] | 10.2 | 69.1 | 0.432 | 0.145 |
| | ReLA [40] | 10.7 | 69.7 | 0.429 | 0.144 |
| | 3D-SPS [41] | 7.9 | 68.8 | 0.402 | 0.158 |
| | IAGNet [12] | 12.9 | 77.8 | 0.443 | 0.129 |
| | LASO [14] | 14.6 | 80.2 | 0.507 | 0.119 |
| | LASO* [14] | 15.6 | 79.9 | 0.549 | 0.108 |
| | 3DAffLLM† [6] | 15.3 | 78.7 | 0.542 | 0.124 |
| | GEAL [5] | 16.7 | 80.9 | 0.567 | 0.106 |
| | **Ours** | **17.1** | **81.5** | **0.571** | **0.103** |

mapping point features to match the LLM token dimension. For the Affordance Decoder, we follow the architecture from LISA [32] but adapted for 3D data. For our iterative self-prompting mechanism, we set the number of refinement iterations $T = 3$ (as performance plateaus beyond this point while computational cost rises sharply (see Figure 3)), with weight parameters $\lambda_t = 0.8^t$ to gradually reduce consistency constraints. In the IDGSP loss, we set $\alpha = 0.1$ for the Tikhonov regularization term. For INAFS, we use $\lambda_1 = 0.5$, $\lambda_2 = 0.3$, and $\beta = 0.05$. The SCSP module uses $K = 3$ frequency bands (following validation in Figure 3) with weights $\gamma_1 = 1.0$, $\gamma_2 = 0.7$, $\gamma_3 = 0.4$, and $\tau = 0.2$ for the total variation term. We use AdamW optimizer with an initial learning rate of $4 \times 10^{-5}$ with cosine scheduling and warm-up ratio of $0.03$. All experiments are done on four NVIDIA V100 GPU with a batch size of 16, training for 20 epochs in $\sim 12hr$.

## 4.3 Findings and Comparison with SoTA

Our proposed framework achieves consistent and substantial performance improvements across the PIAD benchmark, as shown in Table 1, setting a new state-of-the-art for language-guided 3D affordance detection. To ensure a fair and consistent comparison, we reproduced the results of 3D-AffordanceLLM [6] under our evaluation protocol and dataset setup, accounting for differences from the original implementation. In the seen configuration, our method achieves 23.1% aIoU, 85.8% AUC, and 0.664 SIM, outperforming the prior best (GEAL) by relative 2.66%, 0.92%, and 10.48%, respectively. In the unseen split—designed to evaluate zero-shot generalization to novel affordance-object pairs—we obtain 9.2% aIoU, 73.1% AUC, and 0.431 SIM, again surpassing GEAL by 0.5% in aIoU, 0.8% in AUC, and 4.1% in SIM. This improvement reflects our framework's ability to preserve spatial and structural coherence in predicted masks, particularly in ambiguous or underrepresented regions. Earlier fusion-based approaches like [33–38] exhibit significantly inferior performance due to their generic multimodal architectures that fail to model the specialized nature of affordance relationships. These methods are unable

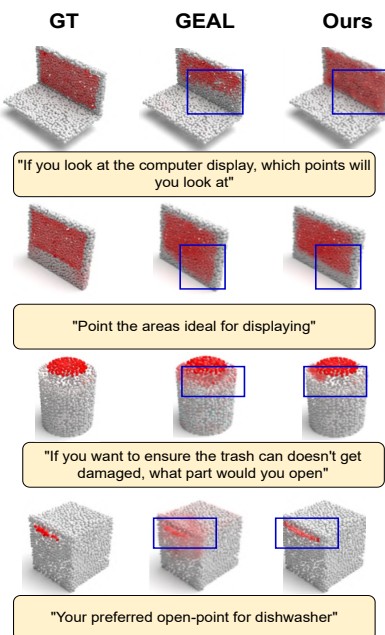

GT GEAL Ours

"If you look at the computer display, which points will you look at"

"Point the areas ideal for displaying"

"If you want to ensure the trash can doesn't get damaged, what part would you open"

"Your preferred open-point for dishwasher"

Figure 4: Qualitative comparison of our affordance segmentation results with GEAL [5].

to bridge the geometric-semantic gap, resulting in
substantial performance degradation (relative aIoU dropping by more than 50% compared to our method). Unlike prior methods such as [14, 6, 5] that operate in a single-pass decoding mode and rely heavily on textual embeddings, our method incorporates closed-loop refinement with geometric feedback, allowing it to resolve fine-grained boundaries in a context-aware manner.

Similar trends are observed on the LASO dataset, where our model achieves 22.8% aIoU, 87.3% AUC, and 0.651 SIM on seen objects, and 17.1% aIoU, 81.5% AUC, and 0.571 SIM in the more challenging unseen setting. Notably, in the unseen split, our framework outperforms the best baseline (GEAL) by 2.4% aIoU, 0.9% AUC, and 0.8% SIM relative. Traditional baselines such as [12, 26, 14] suffer huge degradation in performance when transitioning from seen to unseen due to their rigid, non-adaptive architectures. Even stronger models like [6, 5] exhibit sharp performance drops (e.g., GEAL: 22.0→16.7 aIoU on PIAD), revealing their limited ability to transfer learned affordance priors to unfamiliar topologies.

### 4.4 Ablation Study

The ablation study in Table 2 demonstrates the incremental contribution of each component in our self-prompting framework. When comparing the baseline (row 1) to the full model (row 4), we observe consistent performance improvements across all metrics and datasets, with particularly significant gains in the unseen settings.

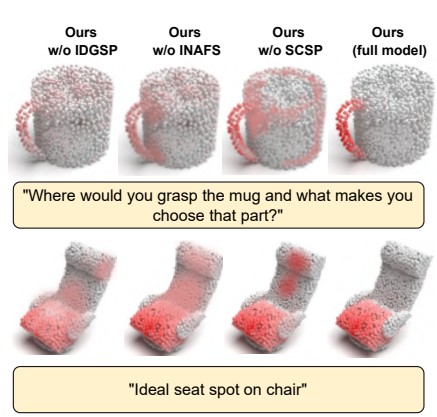

Figure 5: Qualitative visualization of ablation experiments.

**(1)** The *Spectral Convolutional Self-Prompting (SCSP)* module provides the initial performance lift (+0.6/+1.3 aIoU on PIAD/LASO seen), confirming the effectiveness of frequency-domain processing for capturing multi-scale affordance patterns. By operating in the spectral domain, SCSP enables the model to modulate signal components at different structural frequencies, processing both coarse affordance regions and fine boundary details simultaneously (shown in rows 1, 2 of Figure 4). **(2)** Adding *Implicit Neural Affordance Field Self-Supervision (INAFS)* yields further improvements (+0.8/+0.8 aIoU), particularly in structural similarity metrics. This suggests that the continuous implicit field representation enhances boundary precision and produces geometrically coherent affordance regions. The implicit field's ability to capture smooth transitions and model topological relationships proves especially beneficial for complex object geometries, as evident in row 3 of Figure 4. **(3)** The most substantial gains come from incorporating *Iterative Differential Geometry-Based Self-Prompting (IDGSP)*, which provides a significant boost on LASO seen (+2.5 aIoU) and notable improvements across unseen scenarios. This demonstrates that leveraging geometric features (Laplacians, curvatures, normal derivatives) as visual prompts enables the model to progressively refine affordance boundaries through geometric feedback, particularly crucial for distinguishing fine-grained functional regions (row 4 of Figure 4). The synergistic effect of all three components is most pronounced in generalization scenarios, where the relative improvements are larger for unseen settings than seen settings. This confirms our hypothesis that geometric self-prompting enhances the model's ability to adapt to novel object-affordance relationships by leveraging intrinsic geometric cues rather than relying solely on seen training examples. Qualitative visualization is provided in Figure 5. Additional findings can be found in the supplementary file.

### 4.5 Cross-dataset Generalization

Table 3 highlights the cross-dataset generalization performance when models trained on LASO are evaluated on PIAD. Our full model outperforms prior state-of-the-art (GEAL) across all metrics, achieving 19.7%/12.5% aIoU, 84.5%/75.2% AUC, and 0.610/0.465 SIM in seen/unseen splits—marking up to +1.3% aIoU and +0.025 SIM improvements.

Table 2: Ablation study of different components. The best results are in **bold**.

| Type | IDGSP | INAFS | SCSP | PIAD aIoU | AUC | SIM | MAE | LASO aIoU | AUC | SIM | MAE |
|---|---|---|---|---|---|---|---|---|---|---|---|
| Seen | ✗ | ✗ | ✗ | 21.5 | 82.6 | 0.643 | 0.104 | 18.2 | 84.9 | 0.622 | 0.104 |
| | ✗ | ✗ | ✓ | 22.1 | 83.5 | 0.650 | 0.099 | 19.5 | 85.4 | 0.631 | 0.099 |
| | ✗ | ✓ | ✓ | 22.9 | 84.2 | 0.657 | 0.093 | 20.3 | 86.1 | 0.643 | 0.094 |
| | ✓ | ✓ | ✓ | **23.1** | **85.8** | **0.664** | **0.089** | **22.8** | **87.3** | **0.651** | **0.090** |
| Unseen | ✗ | ✗ | ✗ | 7.4 | 71.0 | 0.413 | 0.115 | 15.3 | 78.7 | 0.542 | 0.124 |
| | ✗ | ✗ | ✓ | 8.0 | 72.1 | 0.420 | 0.109 | 16.0 | 79.3 | 0.558 | 0.116 |
| | ✗ | ✓ | ✓ | 8.5 | 72.5 | 0.429 | 0.105 | 16.5 | 80.7 | 0.566 | 0.110 |
| | ✓ | ✓ | ✓ | **9.2** | **73.1** | **0.431** | **0.099** | **17.1** | **81.5** | **0.571** | **0.103** |

The ablation variant without SCSP shows a clear drop (-0.8% seen, -0.7% unseen aIoU), confirming SCSP's role in learning transferable, multi-scale affordance cues. Even without it, our model still outperforms [5, 6], validating the strength of our differential geometry-based self-prompting. All methods exhibit performance degradation in unseen scenarios, but our model maintains the smallest drop, indicating greater robustness to distribution shifts—enabled by shape-aware reasoning rather than dataset-specific memorization.

Table 3: Cross-dataset generalization (LASO→PIAD)

| Method | Seen aIoU | AUC | SIM | Unseen aIoU | AUC | SIM |
|---|---|---|---|---|---|---|
| 3DAffLLM [6] | 17.6 | 82.4 | 0.57 | 10.8 | 72.5 | 0.425 |
| GEAL [5] | 18.4 | 83.2 | 0.59 | 11.6 | 73.8 | 0.44 |
| Ours w/o SCSP | 18.9 | 83.6 | 0.595 | 11.8 | 74 | 0.445 |
| Ours (Full Model) | **19.7** | **84.5** | **0.61** | **12.5** | **75.2** | **0.465** |

## 5 Conclusion and Future Works

We presented *ViSPLA*, a geometry-aware iterative framework for language-guided 3D affordance detection. By combining differential geometric self-prompting, implicit neural fields, and spectral refinement, our model progressively improves affordance segmentation beyond the constraints of single-pass or fixed-label paradigms. Experimental results on LASO and PIAD benchmarks show strong generalization, particularly in zero-shot and cross-dataset settings. Despite its strengths, *ViSPLA* incurs additional computation due to iterative refinement and may face challenges with highly deformable or articulated objects. Future work will explore hybrid prompting strategies that combine geometric and learned latent cues, as well as adaptive iteration control for real-time efficiency. Extending to dynamic scenes and scene-level affordance reasoning is another promising direction. Together, these developments will move us closer to robust, generalizable affordance understanding in complex real-world environments—paving the way for more capable and adaptable embodied agents.

## Acknowledgements

The authors have been supported by the following NSF grants: IIS-2331769, CMMI-2246673, and ECCS-2025929.

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
