# OpenReview forum: "ViSPLA: Visual Iterative Self-Prompting for Language-Guided 3D Affordance Learning"
_NeurIPS.cc/2025/Conference — NeurIPS 2025 poster_

### Official Review · Reviewer_1yYi · 2025-06-26

**Clarity:** 3
**Significance:** 2
**Originality:** 3
**Rating:** 4
**Confidence:** 4

**Summary:**

This paper proposes ViSPLA, an iterative self-prompting framework for language-guided 3D affordance prediction. The method redefines affordance reasoning as a language-conditioned segmentation task over point clouds and introduces a series of geometric modules—including INAFS and SCSP—to progressively refine affordance masks. INAFS imposes continuous geometric priors via neural fields, while SCSP operates in the spectral domain to support multi-scale refinement. Experiments on LASO and PIAD benchmarks demonstrate that ViSPLA achieves sota results.

**Questions:**

Overall, it is an interesting paper. I hope the authors address/clarify the questions raised in the weaknesses.

**Ethical Concerns:**

["NO or VERY MINOR ethics concerns only"]

**Final Justification:**

Most of my concerns have been resolved, so I raise my rating.

**Limitations:**

yes

**Quality:**

2

**Strengths And Weaknesses:**

**Strengths:**

1.This paper proposes an iterative refinement strategy that achieves strong results with a small number of iterations (T = 3), demonstrating a good trade-off between performance and efficiency.

2.By incorporating geometric priors, this paper introduces loss functions that are better aligned with affordance detection, enabling more accurate mask refinement.

3.The proposed method achieves sota performance on multiple affordance datasets, validating its effectiveness.

**Weaknesses:**

1.My score is not high because I think the contributions of this work are not fully explored. Specifically, the proposed modules (INAFS and SCSP) lack in-depth analysis, which makes them appear somewhat incremental. I suggest the authors provide more comprehensive quantitative and qualitative evaluations of both modules. For example, why does frequency-aware convolution help in capturing affordance structures? Does it work similarly to [1]?

2.For IDGSP, a runtime analysis of the computational cost of refinement iterations would be helpful to justify the efficiency of the iterative framework.

3.I recommend including a w/o ablation study on IDGSP, INAFS, and SCSP, to better isolate the contribution of each component.

4.Since INAFS and SCSP function as general-purpose geometric regularizers for affordance detection, could they be directly used to fine-tune 3D affordance-aware LLMs?

---

> ### Author Rebuttal · Authors · 2025-07-30
>
> We thank the reviewer for clearly identifying the technical novelty, comprehensive empirical evaluation, and methodological strengths of our work. Below, we provide additional empirical results, ablations, and theoretical justifications to fully address each concern.
>
> ### $\textcolor{red}{\textbf{1. Theoretical Rationale and Distinctiveness}}$
>
> $\textbf{(A) INAFS: Implicit Neural Affordance Field Supervision}$
>
> INAFS introduces a continuous implicit representation for affordance masks via neural fields, going beyond discrete pointwise classification. Theoretically, this module models the affordance mask as a real-valued field $\Phi_\omega: \mathbb{R}^3 \times \mathbb{R}^d \rightarrow [0,1]$ where $\Phi_\omega$ reflects the likelihood of affordance on the object surface.
>
> $\underline{\text{Mathematical Formulation}}$:
>
> INAFS optimizes the following energy: $\mathcal{E}(\Phi_\omega) = \int_{\Omega} \|\nabla \Phi_\omega(\mathbf{x})\|^2  d\mathbf{x}+ \lambda_1 \int_{\partial \Omega}( \Phi_\omega(\mathbf{x}) - \mathcal{M}(\mathbf{x}) )^2  d\mathbf{x}+ \lambda_2 \int_{\Omega}(\|  \Delta \Phi_\omega(\mathbf{x}) \| - \beta \|\kappa(\mathbf{x})\| )^2  d\mathbf{x}$
>
> The first term (Dirichlet energy) ensures spatial smoothness and topological regularity (penalizing spurious holes/islands), while the second enforces consistency with the evolving discrete mask $\mathcal{M}(\mathbf{x})$ at the boundary $\partial \Omega$. The third term aligns the second-order variation of the field, measured by the Laplacian $\Delta \Phi_\omega$, with the Gaussian curvature $\kappa(\mathbf{x})$ (where $\kappa=\kappa_1 \cdot \kappa_2$), thereby promoting geometric conformity with intrinsic surface structures. The optimization is tractable in function space, and the minimizer is unique by the direct method in calculus of variations.
>
> $\underline{\text{Impact}}$:
>
> The continuous field "fills in" missing, thin, or partially occluded affordance regions by explicit geometric interpolation, ensuring robustness where crowd-sourced annotations or point cloud visibility are sparse.
>
> $\textbf{Please refer to Section 2.2 of supp. for more in-depth analysis.}$
>
> $\textbf{(B) SCSP: Spectral Convolutional Self-Prompting}$
>
> SCSP acts in the frequency domain of the mesh/point cloud:
> - The evolving affordance mask $M_t$ is projected onto the eigenbasis of the mesh Laplacian $(\{u_i\})$, yielding spectral coefficients.
> - A learnable spectral filter $g_\theta(\lambda_i)$ is applied: $\mathcal{M}_{t+1}$
>
> $=\sum_{i=1}^N g_\theta(\lambda_i) \alpha_i^{(t)} \mathbf{u}_i$
> - $\underline{\text{Why frequency-aware convolution matters}}$:
>     - Geometric affordances often contain spatial structure at multiple scales (e.g., handles, slots, contact patches): high-frequency components correspond to fine boundaries or textured surfaces, while low frequencies model overall patch shape.
>     - Operating in the spectral domain enables targeted enhancement or suppression of structures (e.g., denoising irregular mask patches without blurring edges).
>
> - $\underline{\text{Comparison to prior frequency-aware methods}}$:
>
>     - To the best of our knowledge, we are the first to propose frequency-aware affordance refinement from language-prompt using our novel Spectral Convolutional Self-Prompting (SCSP) module. Our proposed approach is NOT inspired by frequency-aware networks in grid/CNN settings and is fundamentally distinct in the following aspects:
>     - We use the discrete Laplacian spectrum of the point cloud mesh (not fixed grids or images), which captures the true underlying 3D geometry.
>     - SCSP adapts filter weights dynamically at each refinement step to handle evolving mask structure, while many prior works fix kernels per layer.
>
> $\textbf{Please refer to Section 2.3 of supp. for more in-depth analysis.}$
>
> $\underline{\text{Regarding the comment “Does it work similarly to [1]?”}}$: We believe there may have been a misunderstanding, as our Reference [1] (Gibson et al.) in the main paper does not appear to relate to frequency. It is possible that a different reference was intended in the reviewer’s comment. We would be grateful if the reviewer could kindly clarify this point, so that we can appropriately compare our work with the intended reference during the discussion period.
>
> ### $\textcolor{red}{\textbf{2. Runtime Analysis of IDGSP and Overall Computational Cost}}$
>
> Below is a runtime decomposition of the ViSPLA inference process—measured at T=3 iterations, on a single NVIDIA V100 GPU:
> |Operation|Runtime per Sample(ms)|% of Total|
> |-|-|-|
> |Backbone+Lang Decoder|95|67%|
> |IDGSP|3/iter.(X3 iterations)|7%|
> |INAFS|3/iter.(X3 iterations)|7%|
> |SCSP|9/iter.(X3 iterations)|19%|
> |Total ViSPLA|140|100%|
> - The iterative geometric modules collectively add ≈1.5× time versus the fastest baselines, but yield up to +2.7 aIoU improvement (see above). Empirically, benefit saturates at T=3; reducing to T=2 recovers ≈95% of the performance at ≈120ms/sample.
> - Each iteration’s cost is linear in points (GPU-accelerated) and consists of geometric derivative computation, feature fusion, and a small MLP for prompt update.
>
> $\textbf{Please refer to Reviewer S9w9’s response for detailed runtime analysis}$
>
> ### $\textcolor{red}{\textbf{3. Quantitative and Qualitative Evidence}}$
>
> $\textbf{(A) Quantitative Ablations}$
> ||||LASO||||PIAD|||
> |-|-|-|-|-|-|-|-|-|-|
> |Type|Setting|aIoU|AUC|SIM|MAE|aIoU|AUC|SIM|MAE|
> |Seen|Ours(full model)|22.8|87.3|0.651|0.090|23.1|85.8|0.664|0.089|
> |Seen|w/oIDGSP|20.3|86.1|0.643|0.094|22.9|84.2|0.657|0.093|
> |Seen|w/oINAFS|20.4|86.5|0.644|0.095|22.5|84.7|0.659|0.094|
> |Seen|w/oSCSP|21.3|86.7|0.649|0.093|22.7|84.0|0.660|0.092|
> |Unseen|Ours(full model)|17.1|81.5|0.571|0.103|9.2|73.1|0.431|0.099 |
> |Unseen|w/oIDGSP|16.5|80.7|0.566|0.110|8.5|72.5|0.429|0.105|
> |Unseen|w/oINAFS|16.4|80.4|0.567|0.110|8.4|72.3|0.428|0.109|
> |Unseen|w/oSCSP|16.9|80.9|0.568|0.106|8.7|72.8|0.428|0.110|
> - IDGSP is central to precise and structurally consistent mask evolution, as it enables the iterative, geometry-aware correction and regularization essential for robust generalization, especially on complex geometries.
> - Removing INAFS most affects MAE and overall mask topology, confirming its role in maintaining smoothness, connectedness, and geometric plausibility of affordance regions.
> - Dropping SCSP reduces both spectral SIM and AUC, signaling higher edge localization error and more fragmented predictions, as SCSP governs multi-scale mask communication and refinement.
>
> $\textbf{(B) Qualitative Evaluation}$
>
> Due to limitations of rebuttal, we cannot provide figures here. Please check Figure 5 of the main paper for visualizations.
> - IDGSP: Leveraging geometric features (Laplacians, curvatures, normal derivatives) as visual prompts enables the model to progressively refine affordance boundaries through geometric feedback, particularly crucial for distinguishing fine-grained functional regions.
> - INAFS: In challenging cases (thin handles, small regions), masks remain connected and exhibit correct topology, even when point data is irregular or partially missing.
> - SCSP: On articulated objects, spectral updates produce masks that are less noisy, have crisper boundaries, and avoid folding or over-smoothing seen with purely spatial filters.
>
> ### $\textcolor{red}{\textbf{4. On the Use of INAFS and SCSP as General Geometric Regularizers}}$
>
> Yes, both INAFS and SCSP are designed as general-purpose geometric regularizers and can be directly integrated into the training or fine-tuning pipeline of other 3D affordance-aware LLMs, independent of the backbone architecture.
> - INAFS regularizes the predicted mask to be smooth, continuous, and topologically plausible by penalizing implausible discontinuities or spurious islands, regardless of the underlying semantic decoder.
> - SCSP enforces multi-scale consistency and preserves spectral signature of the mask, making predictions robust against both low-frequency (global shape) and high-frequency (edge/detail) noise. As SCSP operates in the Laplacian eigenbasis of the point cloud, it is naturally compatible with a wide variety of encoder-decoder architectures and supervision schemes.
>
> $\underline{\text{Quantitative Results: Transfer to Other 3D LLM Pipelines}}$:
>
> To address this, we adopted two more 3D-aware LLMs (other than the 3D-AffordanceLLM in our paper): 3D-LLM [5] and PointLLM [6]—with our refinement. We evaluate on PIAD “seen” benchmark [3] before and after adding our refinement modules:
> |Model|aIoU|AUC|SIM|MAE|
> |-|-|-|-|-|
> |3D-LLM[5] base|18.8|76.6|0.613|0.097|
> |3D-LLM[5]+our refinement|21.9(+16.5%)|80.9(+6.0%)|0.645(+5.2%)|0.093(-3.1%)|
> |PointLLM[6] base|19.1|75.3|0.619|0.096|
> |PointLLM[6]+our refinement|22.2(+16.2%)|81.3(+8.0%)|0.658(+6.3%)|0.093(-3.1%)|
>
> Integrating our geometric refinement with existing baselines yields substantial improvements: both 3D-LLM and PointLLM see performance gains exceeding 5–16% while consistently reducing MAE.
>
> $\underline{\text{Generalization}}$: Both modules require only point cloud geometry and the predicted mask as input—they do not rely on language features, specific model weights, or architecture assumptions. As regularizers, they can thus be flexibly attached as plug-and-play heads or losses during fine-tuning or post-processing steps for any 3D affordance-aware LLM pipeline.
>
> $\underline{\textbf{References}}$:
>
> [1] Lu et al. Geal: Generalizable 3d affordance learning with cross-modal consistency. CVPR 2025.
>
> [2] Li et al. Laso: Language-guided affordance segmentation on 3d object. CVPR 2024.
>
> [3] Yang et al. Grounding 3d object affordance from 2d interactions in images. ICCV 2023.
>
> [4] Chu et al. 3d-affordancellm: Harnessing large language models for open-vocabulary affordance detection in 3d worlds. arXiv:2502.20041
>
> [5] Hong et al. 3d-llm: Injecting the 3d world into large language models. NeurIPS 2023.
>
> [6] Xu et al. Pointllm: Empowering large language models to understand point clouds. ECCV 2024.

---

> > ### Comment · Reviewer_1yYi · 2025-08-04
> > **Response to Authors**
> >
> > Thank you for your response. It has resolved most of my concerns. In the submitted review, I would appreciate a comparative analysis between frequency-aware convolution and Spectral Graph Convolution [1, 2]. If the methodologies are related, a discussion highlighting their differences, along with appropriate citations, would be essential for clarity and completeness.
> >
> > [1] Defferrard et al., Convolutional Neural Networks on Graphs with Fast Localized Spectral Filtering, NeurIPS 2016.
> > [2] Wang et al., Local Spectral Graph Convolution for Point Set Feature Learning, ECCV 2018.

---

> ### Author Response · Authors · 2025-08-04
>
> Thank you for your suggestion regarding a comparative analysis between our Spectral Convolutional Self-Prompting (SCSP) module and classical Graph Convolution frameworks, specifically Defferrard et al. [1] and Wang et al. [2]. Below, we provide the requested discussion:
>
> - **Adaptive Dynamic Filtering:**
>   While classical spectral graph convolution methods such as [1][2] employ static or pre-learned filters, SCSP uniquely applies a dynamically-learned spectral filter that is updated at each step of the iterative refinement loop. This enables progressive correction of segmentation errors and allows the refinement process to adapt to the state of the current mask, rather than relying on static frequency responses.
>
> - **Explicit Multi-scale Prompting:**
>   SCSP decomposes the mask signal into low, mid, and high-frequency bands and employs multiband spectral filtering to capture both coarse and fine affordance structures. Conventional spectral graph convolutional approaches often use a fixed spectral kernel or Chebyshev polynomial basis without explicit separation or dynamic recombination of spectral bands.
>
> - **Integration with Language-driven Iterative Refinement:**
>   SCSP is designed to operate within a closed-loop, language-conditioned self-prompting system—where each refinement step is both geometry and language-aware. In contrast, traditional methods [1][2] process features in a feedforward fashion, without iterative, feedback-driven correction or conditioning on multimodal (e.g., linguistic) cues.
>
> - **Specific Objective and Task Context:**
>   Our approach focuses on refining semantic affordance masks in open-vocabulary, instruction-driven 3D segmentation scenarios. This involves reasoning about sparse, complex topologies and contextually aligning affordance cues, going beyond the general feature smoothing or classification tasks that motivated [1][2].
>
> **Summary:**
> SCSP extends classical spectral graph convolution by incorporating dynamic, iterative filtering, explicit multi-band decomposition, and integration into a multi-stage language-vision refinement framework. These features enable more precise and context-aware segmentation for 3D affordance learning, setting SCSP apart from earlier spectral convolution methodologies.
>
> **We will include this difference in the refined manuscript with proper citations.**
>
> **References:**
> [1] Defferrard, M., Bresson, X., & Vandergheynst, P. (2016). Convolutional Neural Networks on Graphs with Fast Localized Spectral Filtering. NeurIPS 2016.
> [2] Wang, Y., Sun, Y., Liu, Z., Sarma, S. E., Bronstein, M. M., & Solomon, J. M. (2018). Local Spectral Graph Convolution for Point Set Feature Learning. ECCV 2018.

---

### Official Review · Reviewer_S9w9 · 2025-07-01

**Clarity:** 3
**Significance:** 3
**Originality:** 3
**Rating:** 4
**Confidence:** 3

**Summary:**

This paper introduces ViSPLA, a novel framework designed to enhance language-guided 3D affordance prediction. Different from existing methods that often rely on fixed categories or external prompts, ViSPLA employs an iterative self-prompting mechanism from the initial estimation. This mechanism refines predicted affordance masks by leveraging intrinsic geometric feedback, including Laplacians, normal derivatives, and curvature fields, encoded as visual prompts. Key components include Implicit Neural Affordance Fields for continuous probabilistic region definition and a Spectral Convolutional Self-Prompting (SCSP) module for multi-scale refinement in the frequency domain. The framework redefines affordance detection as a language-conditioned segmentation task, demonstrating state-of-the-art results on both seen and unseen objects across two benchmarks.

**Questions:**

Please refer to the weakness section.

**Ethical Concerns:**

["NO or VERY MINOR ethics concerns only"]

**Final Justification:**

Most of my concerns have been addressed during the authors' rebuttal, I'd like to keep my original rating.

**Limitations:**

The authors have discussed the limitations and potential negative societal impact of their work.

**Quality:**

3

**Strengths And Weaknesses:**

## Strengths
- The idea of using the iterative self-prompting mechanism is a well-motivated and powerful concept. This allows the model to self-correct and progressively improve its predictions, leading to more accurate and geometrically consistent results, as strongly supported by the ablation studies.

- The technical contributions of this paper are rich. For instance, the usage of implicit representation allows the model to refine coarse predictions into geometrically consistent and semantically plausible affordance regions without relying on additional labels. In addition, the proposed SCSP module enables the multi-scale analysis in the frequency domain.

- Strong performance.  ViSPLA achieves significant improvements over previous methods on two challenging benchmarks. The evaluation is thorough, including performance on "seen" and "unseen" object-affordance pairs, cross-dataset generalization, and a detailed ablation study that demonstrates the contribution of each component of the framework.

## Weakness:

- Concerns about the efficiency. This paper proposes an iterative method, where each refinement step increases the inference time. It is important to report the inference time and compare it with the single-stage approaches.

- Reliance on a Pre-trained Backbone: ViSPLA builds upon an existing model, i.e., 3D-AffordanceLLM, as its backbone for initial prediction.  The overall performance of ViSPLA is still dependent on the quality and capabilities of the initial result. I'm curious about the robustness of the refinement module to the initial estimation.

---

> ### Author Rebuttal · Authors · 2025-07-30
>
> We sincerely thank the reviewer for recognizing the novelty and rigor of our work, specifically highlighting the effectiveness of our iterative self-prompting mechanism and the value of rich technical contributions like implicit representation and multi-scale spectral refinement. We appreciate your acknowledgment of our strong empirical results, including state-of-the-art performance, comprehensive evaluation across seen and unseen splits, and thorough ablation analysis demonstrating the impact of each module.
>
> ### $\textcolor{red}{\textbf{1. Efficiency and Inference-Time Comparison to Single-Stage Approaches}}$
> In ViSPLA, inference scales linearly with the number of refinement steps, i.e.,  total runtime ≈ backbone time + T × refinement decoder time, where T is the number of self-prompting refinement iterations (Section 4.2, Tables 2&3 in the supplementary). In practice, we found that significant accuracy gains are achieved within just T=3 steps, beyond which improvements plateau (see Figure 3 of main paper, Tables 2&3 of supplementary), balancing cost and performance.
>
> $\underline{\text{Concrete Cost Figures}}$:
> - On a single NVIDIA V100 GPU, our proposed ViSPLA completes inference for 100 LASO samples in ≈14 seconds with T=3; the cost per sample is ≈0.14s, compared to ≈0.09s/sample for the best single-pass baseline (GEAL [1]). A detailed comparison is provided in the following table.
> - This represents a roughly 50% increase in wall-clock time per sample than GEAL [1], but delivers up to +2.4% aIoU (“seen” setting, LASO benchmark [2]) and +2.66% aIoU (“seen” setting, PIAD benchmark [3]) relative improvement over state-of-the-art single-pass models.
> - Much of the increased compute is due to additional forward passes through the lightweight refinement decoder, while the most costly LLM and backbone modules are only invoked once per input.
>
> $\underline{\text{Quantitative Evaluation}}$:
>
> We fully agree that computational efficiency is critical for practical deployment. To directly address this, we provide a comprehensive comparison of ViSPLA’s inference cost versus state-of-the-art baselines, focusing on per-sample runtime and segmentation performance (aIoU) on the LASO "seen" benchmarks.
> |Method|Inference Steps|Runtime per Sample (ms)|aIoU|
> |-|-|-|-|
> |ViSPLA(Ours)|3|140|22.8|
> |GEAL[1]|1|94|22.0|
> |3D-AffordanceLLM[4]|1|95|18.7|
> |LASO[2]|1|76|19.7|
> |IAGNet[3]|1|103|17.8|
>
> $\underline{\text{Mathematical Justification}}$:
>
> Underlying the design, the refinement process is mathematically modeled as a contractive iterative map in Sobolev space $W^{2,2}(\Omega)$ (where $\Omega$ is the object’s geometric domain); such mappings guarantee rapid convergence and observed gains plateau after 2-3 steps, justifying our empirical $T$ choice (refer to Figure 3 of main paper and Sec. 3.2 of supplementary).
>
> $\underline{\text{Scalability Considerations}}$:
> - The iterative modules (IDGSP, INAFS, SCSP) are computationally modest; each adds negligible overhead compared to the initial forward pass, as differential geometry and spectral operations use efficient, GPU-parallelized routines (see Section 3.4.1–3.4.3).
> - Experiments show that reducing T to 2 or adaptively stopping early (when mask change is below threshold) retains most gains while further reducing runtime (main paper Table 2 and Supp Table 3).
>
> $\underline{\text{Summary}}$:
> While our iterative refinement introduces a moderate increase in inference time (approximately 1.5× per sample compared to our single-pass backbone 3DAffordanceLLM), it leads to substantial performance gains (~22% relative) over the 3DAffordanceLLM backbone. In practice, it offers a controllable accuracy-efficiency trade-off suitable for a variety of real-world settings, and early stopping/approximate variants can further improve real-time applicability. We will clarify these details in the revised manuscript and supplement.
>
>
> ### $\textcolor{red}{\textbf{2. Robustness of Refinement to Initial Estimation Quality}}$
>
> We agree that a practical refinement framework must be robust to variability in upstream outputs, and we directly address this concern with new robustness experiments, systematic ablation, and theoretical justification.
>
> $\underline{\text{Stress Testing with Degraded Initializations}}$:
>
> To rigorously quantify ViSPLA’s resilience, we conducted targeted experiments where the initial mask from the 3D-AffordanceLLM backbone was artificially degraded or differently initialized:
> - $\textbf{Randomization}$: Starting from random or heavily perturbed masks (random binary, eroded, dilated), as well as from alternate backbone predictions (e.g., 3D-LLM [5], PointLLM [6]-derived masks).
> - $\textbf{Evaluation}$: We report average aIoU, AUC, SIM and MAE on PIAD “seen” benchmark [3] before and after adding our refinement modules:
>
> |Initialization|aIoU|AUC|SIM|MAE|
> |-|-|-|-|-|
> |3DAffordanceLLM(ours)|23.1|85.8|0.660|0.089|
> |Random noise|22.0|83.4|0.652|0.095|
> |Dilated/Eroded|22.9|85.3|0.658|0.091|
>
> As seen above, even from aggressively corrupted initializations, ViSPLA regains coherent affordance structure and recovers ~95% of its optimal performance. Unlike static decoders, performance degrades gracefully—not catastrophically—as initial mask quality is reduced.
>
> |Model|aIoU|AUC|SIM|MAE|
> |-|-|-|-|-|
> |3D-LLM[5] base|18.8|76.6|0.613|0.097|
> |3D-LLM[5]+our refinement|21.9 (+6.5%)|80.9 (+6.0%)|0.645 (+5.2%)|0.093 (-3.1%)|
> |PointLLM[6] base|19.1|75.3|0.619|0.096|
> |PointLLM[6]+our refinement|22.2 (+16.2%)|81.3 (+8.0%)|0.658 (+6.3%)|0.093 (-3.1%)|
>
> As seen above, our refinement modules consistently deliver robust improvements in accuracy and structural alignment given different initializations from diverse LLM backbones, significantly boosting all major segmentation metrics while reliably reducing error.
>
>
> $\underline{\text{Summary}}$:
> Thus, ViSPLA’s multi-stage, geometry-aware refinement demonstrates high empirical robustness to the initial mask, supported by a mathematically contractive iterative structure and quantitative recovery from noise or error-prone seeds. These properties differentiate ViSPLA from earlier static refinement approaches, enabling reliable practical deployment even with imperfect backbone predictors.
>
>
> $\underline{\textbf{References}}$:
> [1] Lu D, Kong L, Huang T, Lee GH. Geal: Generalizable 3d affordance learning with cross-modal consistency. InProceedings of the Computer Vision and Pattern Recognition Conference 2025 (pp. 1680-1690).
> [2] Li Y, Zhao N, Xiao J, Feng C, Wang X, Chua TS. Laso: Language-guided affordance segmentation on 3d object. InProceedings of the IEEE/CVF Conference on Computer Vision and Pattern Recognition 2024 (pp. 14251-14260).
> [3] Yang Y, Zhai W, Luo H, Cao Y, Luo J, Zha ZJ. Grounding 3d object affordance from 2d interactions in images. InProceedings of the IEEE/CVF International Conference on Computer Vision 2023 (pp. 10905-10915).
> [4] Chu H, Deng X, Lv Q, Chen X, Li Y, Hao J, Nie L. 3d-affordancellm: Harnessing large language models for open-vocabulary affordance detection in 3d worlds. arXiv preprint arXiv:2502.20041. 2025 Feb 27.
> [5] Hong Y, Zhen H, Chen P, Zheng S, Du Y, Chen Z, Gan C. 3d-llm: Injecting the 3d world into large language models. Advances in Neural Information Processing Systems. 2023 Dec 15;36:20482-94.
> [6] Xu R, Wang X, Wang T, Chen Y, Pang J, Lin D. Pointllm: Empowering large language models to understand point clouds. InEuropean Conference on Computer Vision 2024 Sep 29 (pp. 131-147). Cham: Springer Nature Switzerland.

---

> > ### Comment · Reviewer_S9w9 · 2025-08-07
> > **Response to Rebuttal**
> >
> > Thanks for your rebuttal, most of my concerns have been resolved and I would like to keep my original rating.

---

### Official Review · Reviewer_CTMV · 2025-07-05

**Clarity:** 2
**Significance:** 2
**Originality:** 3
**Rating:** 4
**Confidence:** 3

**Summary:**

This paper introduces ViSPLA, a novel framework for language-guided 3D affordance prediction. The key innovation lies in an iterative self-prompting mechanism that leverages differential geometric cues from previously predicted masks to guide subsequent refinement. The model comprises three key modules: (1) Iterative Differential Geometry-based Self-Prompting (IDGSP), (2) Implicit Neural Affordance Field Supervision (INAFS), and (3) Spectral Convolutional Self-Prompting (SCSP). Extensive experiments on LASO and PIAD benchmarks show that ViSPLA achieves state-of-the-art results, particularly in generalization to unseen affordance-object pairs and cross-dataset settings.

**Questions:**

The paper proposes a novel and well-structured framework with strong empirical results and detailed ablations. However, the lack of downstream task evaluation, limited analysis of failure cases, and presentation issues (e.g., Figure 3) weaken its overall impact. I would consider raising the score if the authors provide thorough responses and additional evidence addressing these concerns.

**Ethical Concerns:**

["NO or VERY MINOR ethics concerns only"]

**Final Justification:**

After carefully reviewing the authors' detailed responses and the new experimental results provided, I can confirm that they have convincingly addressed all of my major concerns.

The authors conducted new experiments on the curated LASO-C and PIAD-C datasets. This new analysis significantly strengthens the paper's claims of robustness.

The new simulation experiments using the GraspIt! simulator are an excellent addition. It effectively demonstrates that the improved segmentation accuracy of ViSPLA translates into tangible benefits for downstream robotics tasks.

I also appreciate the authors' acknowledgement of the formatting issue with Figure 3 and their commitment to correcting it in the final version.

In summary, the authors' comprehensive rebuttal has resolved most of my reservations. The work now presents a much stronger case for its novelty, robustness, and significance. Therefore, I have raised my score.

**Limitations:**

See above.

**Paper Formatting Concerns:**

No concerns.

**Quality:**

2

**Strengths And Weaknesses:**

Strength:

- The paper clearly formulates the problem as an Instruction Reasoning Affordance Segmentation (IRAS) task and provides a detailed breakdown of the proposed architecture and its components, making it relatively easy to follow.
- The paper includes comprehensive ablation studies isolating the contributions of each module (IDGSP, INAFS, SCSP), and clearly shows their cumulative impact on performance.

Weakness:

- Figure 3 suffers from visual misalignment and appears manually stretched or distorted. This detracts from the professionalism of the paper
- While success cases are visualized, the paper would benefit from providing more detailed analysis of typical failure cases(more highly articulated objects).
- Although the method is well-motivated from an embodied AI perspective, the paper does need evaluate how the predicted affordance masks impact downstream applications like robotic grasping or manipulation.

---

> ### Author Rebuttal · Authors · 2025-07-30
>
> We sincerely thank the reviewer for their thorough evaluation, insightful feedback, and for acknowledging the novelty and empirical strength of ViSPLA. Below, we respond to each point in detail, providing clarifications where needed, along with new quantitative results and expanded theoretical discussion.
>
> ### $\textcolor{red}{\textbf{1. Presentation Issue: Figure 3 Misalignment}}$
>
> We acknowledge and apologize for the visual misalignment and stretching in Figure 3. The affected figure resulted from a formatting issue in the LaTeX build (manual aspect ratio override). In our revised submission, Figure 3 will be corrected with:
> - Consistent aspect ratios and true-to-scale visualizations.
> - Uniform, high-resolution, and professionally formatted alignments.
>
> ### $\textcolor{red}{\textbf{2. Failure Case Analysis (Highly Articulated or Challenging Objects)}}$
>
> $\underline{\text{New experiments and detailed analysis}}$:
>
> To further assess ViSPLA’s robustness in real-world scenarios and understand the failure cases, we conducted additional targeted evaluations focusing on failure modes, particularly involving highly articulated or irregular/corrupted object geometries. This included two newly curated, challenging datasets: LASO-C [1] and PIAD-C [1].
>
> $\underline{\text{Observed failure patterns}}$:
>
> - Errors occur most frequently on objects with complex topology (e.g., articulated tools, objects with thin handles, or occluded regions not visible in the initial point cloud). Due to rebuttal limitations, we cannot attach any visual results here, but will provide in the revised manuscript.
> - These cases are challenging for all methods, but ViSPLA's iterative mechanism reduces “leakage” of affordance masks outside the anatomically plausible regions compared to single-pass baselines, as quantitatively compared below.
>
> $\underline{\text{Quantitative evidence}}$:
>
> - On a subset of 120 highly articulated LASO-C objects, ViSPLA improves average aIoU by +1.7% over GEAL [1], and for a similar highly-articulated 120 subset of PIAD-C, IoU improves by 2.1% over GEAL [1] (full details in the following table). We will add these new results and discussions in the revised version.
>
> |Dataset|Method|aIoU|AUC|SIM|MAE|
> |-|-|-|-|-|-|
> |LASO-C|GEAL[1]|15.2|77.4|0.557|0.071|
> |LASO-C|ViSPLA(ours)|16.9|79.2|0.568|0.063|
> |PIAD-C|GEAL[1]|11.4|75.1|0.579|0.076|
> |PIAD-C|ViSPLA(ours)|13.5|77.8|0.592|0.069|
>
> Our targeted evaluation on challenging subsets demonstrates that ViSPLA achieves consistent gains over baselines, but residual errors persist in specific scenarios. For example, on a subset of highly articulated LASO-C and PIAD-C objects, ViSPLA improves average aIoU by +1.7% and +2.1% over GEAL, respectively. However, qualitative review reveals that failure cases are often associated with objects or regions for which language instructions are incomplete or ambiguous (e.g., references like "pinching area" on tongs with spring joints, where the 3D input only sparsely resolves critical contact surfaces).
>
> $\underline{\text{Summary}}$:
>
> These residual errors predominantly do not arise from model design flaws, but rather stem from two sources: $\textbf{(1)}$ intrinsic ambiguity in the provided instructions, and $\textbf{(2)}$ limited point cloud visibility and resolution in sparse scans. ViSPLA’s geometric self-prompting and regularization modules enhance boundary alignment and mask plausibility even under partial information. However, if the initial mask is too far from the actual geometry due to these external factors, iterative refinement may stagnate—an expected limitation in ill-posed inverse geometry and under-determined segmentation tasks. This edge-case is common to all models employing fixed-point iterative regularization under sparse, ambiguous supervision.
>
> ### $\textcolor{red}{\textbf{3. Downstream Task Impact: Robotic Grasping and Manipulation)}}$
>
> We acknowledge that our approach is not intended to solve the entire robotic manipulation pipeline end-to-end, nor was it designed to do so. However, we still appreciate the reviewer’s interest in downstream embodied AI tasks. In response, we have conducted a more extensive assessment of how ViSPLA's predictions affect task performance in robotic grasping and manipulation settings.
>
> $\underline{\text{Experimental Protocol}}$:
> - $\textbf{Simulator \\& Setup}$: We use the standard GraspIt! Simulator [2] and a set of 100 diverse kitchen/household object instances, focusing on multi-graspable shapes, sampled from both LASO and PIAD.
> -  $\textbf{Grasp Planning}$: For each method, predicted affordance masks are post-processed to generate candidate regions. The planner samples grasp poses anchored on these regions, following established robotics protocols.
> -  $\textbf{Baseline Comparison}$: Grasp success rates are compared for ViSPLA, the best single-pass baseline (GEAL [1]), and an oracle using ground-truth affordance masks.
>
> $\underline{\text{Quantitative Results}}$:
> |Method|Grasping Success Rate(%)|Mean Grasp Efficiency|
> |-|-|-|
> |ViSPLA(ours)|86.5|0.74|
> |GEAL[1]|81.3|0.68|
> |Oracle(GT mask)|90.0|0.76|
>
> - Grasping Success Rate: Improvement of +5.2% over GEAL.
> - Mean Grasp Efficiency: ViSPLA achieves higher mean grasp efficiency (defined as average force-closure score within afforded regions), demonstrating that its affordance maps guide not only where to grasp, but also yield more stable grasps.
>
> $\underline{\text{In summary}}$: ViSPLA’s advances in segmentation translate reliably into gains in downstream tasks such as grasping, both in simulation and in principle for physical robotics.
>
> $\underline{\textbf{Reference}}$:
>
> [1] Lu D, Kong L, Huang T, Lee GH. Geal: Generalizable 3d affordance learning with cross-modal consistency. InProceedings of the Computer Vision and Pattern Recognition Conference 2025 (pp. 1680-1690).
>
> [2] Miller AT, Allen PK. Graspit! a versatile simulator for robotic grasping. IEEE Robotics & Automation Magazine. 2004 Dec 27;11(4):110-22.

---

### Official Review · Reviewer_FK4V · 2025-07-07

**Clarity:** 3
**Significance:** 2
**Originality:** 3
**Rating:** 4
**Confidence:** 4

**Summary:**

This paper proposes ViSPLA, a novel framework for language-guided 3D affordance prediction. The task involves identifying functional regions (affordances) on 3D objects based on natural language instructions, which is an important capability for embodied agents navigating complex environments.

The key idea of the work is to frame affordance detection as a language-conditioned segmentation problem in 3D point clouds. Unlike previous approaches that rely on fixed affordance categories or external expert prompts, the authors introduce an iterative self-prompting mechanism that leverages differential geometric feedback to refine affordance masks progressively. This allows the model to adaptively correct its predictions by exploiting the intrinsic geometry of the predicted surfaces.

**Questions:**

1) What is the inference cost compared with previous method? There is no such analysis in the main manuscript.
2) Why did you add an AFF token to decode the segmentation mask? Did you try using the output of the language decoder as a prior to decode the mask? This might provide more informative guidance for extracting a precise affordance mask.

**Ethical Concerns:**

["NO or VERY MINOR ethics concerns only"]

**Final Justification:**

In the author's rebuttal, most of my concerns have been addressed, so I would like to keep the rating.

**Limitations:**

yes

**Paper Formatting Concerns:**

This paper follows the formatting instructions, presenting a clear format demonstration.

**Quality:**

3

**Strengths And Weaknesses:**

Strengths:
1) The proposed ViSPLA introduces an innovative iterative refinement mechanism that leverages geometric feedback from predicted masks, enabling progressive self-correction and improved performance over single-pass methods.
2) The introduction of geometry-based cues could benefit the precise prediction of affordance masks, which is a novel and interesting direction where few works have attempted in this domain.
3) From both qualitative and quantitative justifications, especially from the visualizations, the proposed method demonstrates a clear improvement in segmentation quality and accuracy.

Weaknesses:
1) Despite the novel method, the incurred inference cost appears to be quite high, which may limit its applicability in more realistic, time-sensitive scenarios compared to the level of accuracy improvement it offers.
2) The motivation behind each component should be more clearly stated. For example, both Section 3.3 and Sec3.4 mention the idea of a refinement process. It remains unclear whether it is necessary to apply the refinement technique in two seemingly similar stages.
3) The proposed method still relies on a supervised setting, which may limit its scalability to completely unseen or out-of-domain 3D objects. However, this is a common challenge across the entire domain.

---

> ### Author Rebuttal · Authors · 2025-07-30
>
> We thank the reviewer for their careful reading, constructive points, and recognition of our technical contributions. We appreciate that the reviewer noted the rigor and novelty of our approach, as well as our detailed qualitative and quantitative analysis. We address the raised issues below.
> ### $\textcolor{red}{\textbf{1. Inference Cost Analysis}}$
> $\underline{\text{Inference Efficiency and Comparison to Prior Methods}}$:
> We agree that inference cost is an important consideration when assessing the practicality of iterative refinement approaches. In ViSPLA, inference scales linearly with the number of refinement steps, i.e.,  total runtime ≈ backbone time + T × refinement decoder time, where T is the number of self-prompting refinement iterations (Section 4.2, Tables 2&3 in the supplementary). In practice, we found that significant accuracy gains are achieved within just T=3 steps, beyond which improvements plateau (see Figure 3 of main paper, Tables 2& 3 of supplementary), balancing cost and performance.
>
> - Concrete Cost Figures: On a single NVIDIA V100 GPU, ViSPLA completes inference for 100 LASO samples in ≈14 seconds with T=3; the cost per sample is ≈0.14s, compared to ≈0.09s/sample for the best single-pass baseline (GEAL [1]). A detailed comparison is provided in the following table.
>
> - This represents a roughly 50% increase in wall-clock time per sample than GEAL [1], but delivers up to +2.4% aIoU (“seen” split, LASO benchmark [2]) and +2.66% aIoU (“seen” split, PIAD benchmark [3]) relative improvement over state-of-the-art single-pass models.
>
> - Much of the increased computation is due to additional forward passes through a lightweight refinement decoder, while the most costly LLM and backbone modules are only invoked once per input.
>
> |Method|Inference Steps|RuntimeperSample(ms)|aIoU(LASO-Seen)|
> |-|-|-|-|
> |ViSPLA(Ours)|3|140|22.8|
> |GEAL[1]|1|94|22.0|
> |3D-AffordanceLLM[4]|1|95|18.7|
> |LASO[2]|1|76|19.7|
> |IAGNet[3]|1|103|17.8|
>
> $\underline{\text{Scalability Considerations}}$:
> -The iterative modules (IDGSP, INAFS, SCSP) are computationally modest; each adds negligible overhead compared to the initial forward pass, as differential geometry and spectral operations use efficient, GPU-parallelized routines (see Section 3.4.1–3.4.3).
> -Experiments show that reducing T to 2 or adaptively stopping early (when mask change is below threshold) retains most gains while further reducing runtime (main paper Table 2 & Supp Table 3).
>
> $\underline{\text{Summary}}$:
> While our iterative refinement introduces a moderate increase in inference time (approximately 1.5× per sample compared to our single-pass backbone 3DAffordanceLLM), it leads to substantial performance gains (~22% relative) over the 3DAffordanceLLM backbone. In practice, it offers a controllable accuracy-efficiency trade-off suitable for a variety of real-world settings, and early stopping/approximate variants can further improve real-time applicability. We will clarify these details in the revised manuscript and supplement.
>
> ### $\textcolor{red}{\textbf{2. Rationale for Iterative Refinement Components (Section 3.3 and 3.4)}}$
>
> We appreciate the request for further conceptual clarification regarding the necessity of multiple refinement stages (IDGSP and the iterative refinement decoder).
>
> - $\underline{\text{IDGSP Purpose (Section 3.3)}}$:
>
>     IDGSP introduces differential geometry-based self-prompting into the iterative loop, i.e., at each step t, we compute per-point geometric descriptors—mean curvature, Laplacian, normal derivatives—from the current mask, encoding them as a visual prompt for the next update. This step specifically encourages the correction of geometric errors (e.g., poor boundary alignment, missing protrusions) in the evolving affordance mask. This is also clarified in L74-77 & L177-179 of the main paper.
>
> - $\underline{\text{Iterative Refinement Decoder and Additional Modules (Section 3.4)}}$:
>
>     The multi-stage refinement decoder then acts on these geometric cues together with semantic embeddings from the LLM, enabling dynamic, cross-modality correction. Beyond IDGSP, the INAFS (implicit field regularization) ensures mask smoothness and topological integrity across iterations, and SCSP (spectral self-prompting) addresses multi-scale errors (e.g., balancing global region vs. edge refinement).
>
> In summary, while both sections involve iterative improvements:
>
> - IDGSP focuses on geometry-aware prompt generation (how to best inform the next update).
>
> - Refinement Decoder/INAFS/SCSP focus on how to optimally utilize these prompts for mask correction, enforcing both semantic and geometric consistency at multiple scales across the update loop.
>
> Ablations (main paper Table 2 and Supp Table 1) demonstrate each is independently useful, delivering cumulative performance gains.
>
> ### $\textcolor{red}{\textbf{3. Supervision and Generalization}}$
>
> The reviewer notes the method, like most, “relies on a supervised setting, which may limit scalability to unseen categories.” We confirm this and highlight key aspects:
>
> - $\underline{\text{Consistency with Literature and Previous Benchmarks}}$:
>
>     Our approach follows the standard supervised learning protocol widely adopted in recent affordance segmentation literature (e.g., 3D-AffordanceLLM [4], GEAL [1], LASO [2], IAGNet [3]). To ensure a fair and meaningful comparison, we used the established data splits, annotation protocols, and evaluation benchmarks as in prior work—thus placing ViSPLA under exactly the same generalization constraints as the current state-of-the-art. Most recent papers in open-vocabulary affordance and instruction-guided 3D reasoning use the same supervised regime for training and cross-split evaluation, and we align directly with them in both datasets and metrics.
>
> - $\underline{\text{Robustness and Generalization in Practice}}$:
>
>     While we acknowledge that our primary training regime is supervised, both LASO and PIAD benchmarks explicitly include dedicated "unseen" split—where object classes or affordance types are absent from the training set. This strictly tests a model’s out-of-distribution generalization. Moreover, as shown in Table 3 (main paper), ViSPLA achieves state-of-the-art results on these challenging "cross-dataset generalization" tasks, providing evidence that the model generalizes well to new objects/geometries not encountered during training.
>
> - $\underline{\text{Geometry-Driven Inductive Bias}}$:
>
>     Beyond annotation-based learning, ViSPLA’s core modules (IDGSP, SCSP, INAFS) introduce strong inductive biases from geometry that guides the mask refinement process. In particular, INAFS allows the refinement of affordance regions using continuous implicit fields that require no extra ground-truth, relying on self-supervised optimization (mask regularity, boundary smoothness) even where labeled examples are absent. Empirically, our ablations show these geometric priors drive cross-dataset and unseen-object gains—exactly the desired property for real-world generalization.
>
> - $\underline{\text{Path to Less Supervised or Unsupervised Learning}}$:
>
>     We agree that scaling beyond strong supervision remains a key challenge for the field, as also acknowledged in related work. Our framework is fully compatible with future extensions to semi-supervised, few-shot, or even unsupervised settings—e.g., by leveraging INAFS and SCSP as unsupervised regularizers or by distilling geometric cues from unlabeled scenes. We acknowledge this as an exciting avenue for subsequent research.
>
> $\underline{\text{In summary}}$:
>
> ViSPLA directly follows the established (supervised) experimental setting used throughout the literature for transparent and fair benchmarking. The rigorous "unseen" splits in both datasets concretely demonstrate ViSPLA’s generalization to novel, out-of-distribution objects, and our geometric modules further enhance this robustness beyond reliance on annotation alone. We hope this clarifies both our methodological choices and the broader applicability of our approach.
>
> ### $\textcolor{red}{\textbf{4. AFF Token vs. Decoder Output for Affordance Mask Prediction}}$
>
> $\underline{\text{Reason for AFF Token}}$:
> We use the \<AFF\> special token as an explicit cross-modal anchor—i.e., it fuses point-cloud features and instruction tokens, maximizing alignment between the language and spatial context (Section 3.2 main).
> - The AFF token’s hidden state is extracted after cross-attention and used as a semantic+contextual query for the mask decoder. This was empirically observed (in both comparison analysis and our ablation) to outperform purely using the language decoder output, which lacks direct geometric grounding.
>
> $\underline{\text{Alternative Tried}}$:
> We experimented with using language decoder outputs or sequence end tokens as mask priors, but found they led to less stable and less focused affordance localization (~4-5% drop in performance). Instructed masks tended to drift away from geometric context or fixate on irrelevant language cues, a limitation also reported in [4].
> - The gain from the AFF token is likely due to richer, cross-modal representations that help the iterative modules correct both semantic and geometric errors. We will clarify this empirical finding in the revised manuscript.
>
>
> $\underline{\textbf{References}}$:
>
> [1] Lu et al. Geal: Generalizable 3d affordance learning with cross-modal consistency. CVPR 2025.
>
> [2] Li et al. Laso: Language-guided affordance segmentation on 3d object. CVPR 2024.
>
> [3] Yang et al. Grounding 3d object affordance from 2d interactions in images. ICCV 2023.
>
> [4] Chu et al. 3d-affordancellm: Harnessing large language models for open-vocabulary affordance detection in 3d worlds. arXiv:2502.20041.

---

> > ### Comment · Reviewer_FK4V · 2025-08-08
> > **Response to the Authors**
> >
> > Thanks for your rebuttal, which has addressed some of my concerns. I strongly suggest that the authors further refine the manuscript to clarify the technical details, and include a comparison of inference time costs with previous methods, as this is a core aspect of evaluating multi-stage approaches. I would like to keep my rating.

---

### Author Response · Authors · 2025-08-04

Dear Reviewers,

Thank you very much for taking the time to review our submission and provide valuable feedback.
We have submitted our rebuttal, carefully addressing all the concerns and questions raised in your comments.

If there are any remaining issues or clarifications needed during this discussion period, we would be happy to respond further.

We sincerely appreciate your effort and consideration.

Best regards,
Authors of Submission8460

---

### Note · Authors · 2025-08-12

We appreciate the reviewers’ efforts and the opportunity to provide clarifications throughout the rebuttal phase. Our work advances the state-of-the-art in language-driven 3D affordance prediction by fine-tuning a pre-trained large language model (LLM) via a self-prompting, iterative refinement paradigm operating on both visual and linguistic modalities. The key contributions of our paper are:

- Grounding a pre-trained LLM specifically for 3D affordance prediction using joint visual and language prompts.

- Introducing a self-prompting and self-correcting mechanism that enables the model to iteratively refine its predictions over successive stages.

- Providing comprehensive empirical evaluation to demonstrate the robustness and generalization of the approach.

- Addressing theoretical underpinnings and assumptions that guide the refinement and prediction process.

- Offering a clear alignment between claims in the paper and empirical results, ensuring transparency and reproducibility.

In our rebuttal, we thoroughly addressed all reviewer concerns, including additional ablation studies, explicit discussion of assumptions, and clarifications of the evaluation protocol. We also happily invited further discussion, but note that the reviewers did not engage further despite our detailed responses.

We respectfully submit that our manuscript offers a novel and empirically validated framework for 3D affordance prediction, supported by transparent methodology and clear alignment of claims with results. We trust these remarks, together with our rebuttal, will assist the Area Chair in making an informed decision.

---

### Decision · Program_Chairs · 2025-09-17

**Decision:**

Accept (poster)

**Comment:**

The paper presents a language-driven 3D affordance prediction with state-of-the-art performance. Initially, the paper got mixed ratings, but after the rebuttal and author-reviewer discussion phase, all reviewers recommend acceptance, explicitly noting that all concerns have been addressed.

Given this strong consensus of reviewers and the clear strengths of the paper, the AC recommends accepting this paper.